# Individual Trajectories of Bone Mineral Density Reveal Persistent Bone Loss in Bone Sarcoma Patients: A Retrospective Study

**DOI:** 10.3390/jcm11185412

**Published:** 2022-09-15

**Authors:** Sofia Avnet, Luigi Falzetti, Alberto Bazzocchi, Chiara Gasperini, Fulvia Taddei, Enrico Schileo, Nicola Baldini

**Affiliations:** 1Department of Biomedical and Neuromotor Sciences, University of Bologna, 40127 Bologna, Italy; 2Biomedical Science and Technologies and Nanobiotechnology Laboratory, IRCCS Istituto Ortopedico Rizzoli, 40136 Bologna, Italy; 3Diagnostic and Interventional Radiology, IRCCS Istituto Ortopedico Rizzoli, 40136 Bologna, Italy; 4BIC Bioengineering and Computing Laboratory, IRCCS Istituto Ortopedico Rizzoli, 40136 Bologna, Italy

**Keywords:** bone sarcoma, BMD, computed tomography, chemotherapy, cancer survivors, osteopenia

## Abstract

Multiagent chemotherapy offers an undoubted therapeutic benefit to cancer patients, but is also associated with chronic complications in survivors. Osteoporosis affects the quality of life of oncologic patients, especially at the paediatric age. However, very few studies have described the extent of loss of bone mineral density (BMD) in bone sarcoma patients. We analysed a retrospective series of children and adolescents with primary malignant bone tumours (52 osteosarcoma and 31 Ewing sarcoma) and retrieved their BMD at diagnosis and follow-up as Hounsfield units (HU). We studied their individual BMD trajectories before and after chemotherapy up to 5 years, using routine chest CT scan and attenuation thresholds on T12 vertebrae ROI. At one year, bone sarcoma patients showed significant bone loss compared to diagnosis: 17.6% and 17.1% less for OS and EW, respectively. Furthermore, a bone loss of more than 49.2 HU at one-year follow-up was predictive of the persistence of a reduced bone mass over the following 4 years, especially in patients with EW. At 4 years, only 26% and 12.5% of OS and EW, respectively, had recovered or improved their BMD with respect to the onset, suggesting a risk of developing morbidities related to a low BMD in those subjects.

## 1. Introduction

Standard chemotherapy offers an important therapeutic benefit in cancer patients, but at the same time can be responsible for late complications in survivors, requiring additional therapies or preventive strategies. Among the long-term medication-induced adverse effects, a reduction in bone mineral density (BMD), known as secondary osteoporosis, may be detected [1,2]. In principle, this problem is even more relevant in paediatric and young adults, due to their expected longer life expectancy [3,4].

Osteosarcoma (OS) is the most frequent primary malignant bone tumour (8–11/million/year) with a peak incidence at 15–19 years of age [5]. Ewing sarcoma (EW), the second most frequent bone sarcoma (1.5/million/year), has a similar age incidence [6]. In both cases, the standard of care consists of a multimodal regimen including intensive multiagent chemotherapy in addition to surgery and/or radiotherapy [7]. Anticancer drug regimens induce persistent toxicity on the growing skeleton by compromising the achievement of a satisfactory peak bone mass, which is crucial to achieve an adequate BMD in late adulthood [8,9]. Although evidence of specific effects of single drugs on BMD loss is lacking [10], previous authors have alarmingly reported significantly decreased BMD values based on the analysis of bone health status by dual-energy X-ray absorptiometry (DXA), both after the completion of neoadjuvant chemotherapy and at clinical remission [11,12,13,14,15]. Specifically, in three different studies, at 6, 7, 8, and 10 years from diagnosis, around 87%, 58%, and 65% of patients, respectively, were osteopenic or osteoporotic, or showed a low BMD Z-score [13,14,15]. Similar results were obtained using quantitative computed tomography (CT). Kaste et al. found a reduction in BMD status with a median Z-score of −0.75 in a heterogeneous population of 99 paediatric cancer survivors, including 24 OS and 25 EW, at least one year after the completion of chemotherapy [10].

Although there is solid evidence on the development of secondary osteoporosis in long-term survivors of bone sarcomas, this is still a neglected field of investigation [16,17]. In particular, it has not been established yet how to identify patients in whom BMD is permanently compromised [18]. A number of factors may be involved, such as the use of methotrexate or cyclophosphamide in the treatment protocol, treatment duration, or other factors that may affect the extent of long-term toxicity of chemotherapy, such as the presence of vitamin D deficiency, age and weight at diagnosis, low calcium intake, endocrinopathies, genetically determined lactose intolerance, or tubular renal dysfunction [10,19]. However, results obtained from these studies are often contradictory, making it impossible at present to define patients at risk of developing early osteoporosis, and who may benefit from timely interventions to prevent secondary osteoporosis [12,14,17]. 

In 2018, the Children’s Oncology Group (COG) published guidelines for long-term follow-up of childhood, adolescent, and young cancer survivors, recommending BMD assessment by DXA at 2 ys after completion of therapy only for patients treated with methotrexate [20]. However, the COG also pointed out the need for additional studies to include more homogeneous patient groups in order to identify conditions associated with a decreased BMD. In our opinion, these could be ascertained only by an accurate comparison of bone health at follow-up and diagnosis in the same subject. Indeed, given the interpatient variability of BMD and the multitude of predisposing factors, only an analysis based on individual trajectory has the potential to add significant information. 

In this study, we opportunistically analysed a retrospective cases series of primary bone tumours (OS and ES) for whom BMD data were retrieved at the clinical onset, during and after treatment up to 5 years after the end of therapy, thus analysing individual trajectories in a thorough manner in order to ascertain the effect of chemotherapy on bone health and to identify patients at higher risk of developing secondary osteoporosis.

## 2. Subjects and Methods

### 2.1. Patient Information and Data Collection

We retrospectively analysed a continuous series of 168 patients with OS and 88 with EW, treated at our Institution between May 2007 and December 2009. Patients here considered were not treated with the randomized EuroEWING2012 (clinicaltrialsregister.eu, EudraCT Number: 2012-002107-17) and OS2006 (clinicaltrials.gov, NCT number: NCT00470223, accessed on 6 August 2022) protocols in which zoledronate was evaluated for its presumptive anticancer activity. All patients underwent routine chest CT scan at diagnosis and during their follow-up for staging. Inclusion criteria: patients were examined with the same CT scanner (BrightSpeed series, GE Healthcare Waukesha, Wisconsin) and the same acquisition protocol (120 kV scan and slice thickness of 3.75 mm) at diagnosis and during the subsequent 5 y follow-up. Exclusion criteria: age over 45 ys, presence of other conditions that could affect bone metabolism, diseases involving the spine or conditions that could cause significant CT image artefacts (i.e., metallic devices). Based on the exclusion and inclusion criteria, 52 patients with OS and 31 with EW were included from those initially considered. Six age-matched children and adolescents undergoing CT examination for spine trauma in our hospital were considered as control. The study was approved by the local Ethics Committee (CE-AVEC 434/2019/Oss/IOR). 

OS patients received standard treatment consisting of preoperative multiagent chemotherapy, surgical excision, and postoperative chemotherapy according to standard protocols. EW patients were treated with the combination of pre- and postoperative chemotherapy plus radiotherapy vs. surgery as local control. Fourteen high-risk EW patients received high-dose chemotherapy (HDCT) followed by autologous peripheral blood stem cell transplantation (ASCT) for disease relapse or progression. The patient characteristics are summarized in Table 1. We retrospectively retrieved laboratory results, chest CT imaging, and other clinical data from medical records. Laboratory parameters, including C-reactive protein (CRP), alkaline phosphatase (ALP), and lactate dehydrogenase (LDH) were routinely assessed at diagnosis. 

### 2.2. Measurements of Bone Mineral Density

To retrospectively obtain a surrogate measure of trabecular BMD, we used a method based on the measurement of a region of interest (ROI) of the acquired CT scan. This method has been validated by the comparison with DXA T-score for the assessment of osteoporosis [19]. We used attenuation levels as indicative of trabecular BMD, expressed as Hounsfield units (HU) and assessed on T12 vertebrae ROI, at diagnosis and then annually for 5 years. For image analysis, we used PACS software Carestream VUE version 11.4.1.1011 (Carestream health, Rochester, New York, NY, USA). First, we excluded the presence of vertebral abnormalities (i.e., compression fractures) using sagittal chest CT views of the lumbar spine. Next, we identified the T12 vertebra as the last rib-bearing vertebra. In the axial view of the chest CT, a single oval ROI was placed with click-and-drag on an area of trabecular bone of the vertebral body at the central portion of the vertebrae. The ROI was placed on the anterior portion of the vertebrae to avoid the venous plexus, the cortical bone, and any vertebral defect [19].

### 2.3. Statistics

Statistical analyses were performed with GraphPad Prism 7 (GraphPad Software, San Diego, CA, USA) for all but one test: for the ROC curve analysis, we used MedCalc Software (version 7.5.0.0, Mariakerke, Belgium). Due to the small number of observations, data were not considered as normally distributed and nonparametric tests were used. To test the difference between two groups we used the Mann–Whitney U test (unpaired analysis), and the Wilcoxon test (paired analysis). To test correlation, the Spearman rank test (two-tailed) was used. For the evaluation of the association between BMD ROI attenuation values at T0 and the risk of relapse, we considered only a subgroup of patients without the presence of clinically evident metastases at diagnosis (classical OS and EW). For the analysis of sensitivity and specificity in predicting the occurrence of long-term BMD decreased in bone sarcoma survivors and for plotting the receiver operating characteristic (ROC) curve, we used the 70th percentile of the differences between the CT attenuation value at T0 in respect to T1 (Δ_T0–T1_) of the combined populations of OS and EW patients as a cut-off value. This value corresponded to −49.23 HU. An AUC above 0.8 provides convincing prognostic evidence (excellent when above 0.9), while those below 0.8 provide fair/bad prognostic evidence [21,22]. The clinical usefulness of the test was also expressed by the likelihood ratio values: positive and negative likelihood ratios state how many times more likely it is that a measured parameter will change in patients with disease than in those without a disease. A positive likelihood ratio above 10 and a negative likelihood ratio below 0.1 provide convincing diagnostic evidence, while those above 5 and below 0.2 provide strong prognostic evidence [23]. Data were expressed as median unless otherwise specified, and only *p*-values < 0.05 were considered statistically significant. 

## 3. Results

### 3.1. Characteristics of the Studied Populations

Consistent with the previous literature, in our series the peak of incidence for both OS and EW was between 11 and 20 years of age.

The median age was slightly lower in patients with OS than in those with EW (Table 1). OS involved the limbs in 94.2% of cases and the axial skeleton in 5.8% of cases, while EW occurred in limbs in 67.7% of cases and the axial skeleton, but not the spine, in 32.3% of cases (Table 1). 

### 3.2. BMD at the Onset

At diagnosis, the median ROI attenuation was 243.31 HU (min–max 149.79–316.56 HU) for OS and 225.65 HU (min–max 158.13–321.31 HU) for EW. We found no significant difference in BMD between OS and EW groups, nor between cancer patients and controls (Figure 1A). 

Furthermore, we found no correlation between BMD and tumour localization (data not shown), or with gender in EW patients, whereas in OS females showed a higher BMD than males (Figure 1B, *p* = 0.0433). In our series, 17% of OS and 23% of EW patients had clinically evident lung metastases already at the disease onset (Table 1). However, the presence of metastases did not affect BMD (Figure 1C). BMD also did not correlate with prognosis, as revealed by the analysis of relapsed versus non-relapsed patients in the classic OS subgroup (without metastases at diagnosis, Figure 1D). Similar results were obtained with predictive biochemical markers such as LDH and ALP [24], which did not correlate with BMD at diagnosis. The only parameter that was significantly and inversely correlated with a lower CT ROI attenuation was CRP, and only in patients with OS (Figure 1E, *p* = 0.0219).

### 3.3. BMD at the Completion of Chemotherapy

Already after around 1 year from the onset (T1), upon completion of the treatment protocol, both OS and EW patients showed a significant reduction in BMD, as assessed by the attenuation of CT ROI (median 243.3 HU at T0 vs. 200.1 HU at T1 for OS, and 225.7 HU at T0 vs. 187.4 HU at T1 for EW, **** *p* < 0.0001 with Wilcoxon test paired analysis for both types of sarcomas). In Figure 2A,B is also showed the results of unpaired analysis (Mann–Whitney U test). However, no significant difference in the magnitude of BMD reduction (Δ_T1–T0_) was detected between OS and EW patients.

Specifically, OS patients showed a median decrease in bone density of 17.7% and 16.9% for EW. Furthermore, only in OS patients, the presence of lung metastases at the onset was associated with a greater loss of the bone mineral component (Figure 2C, *p* = 0.0203). Finally, the extent of BMD loss immediately after treatment was significantly and inversely correlated with patient age (Figure 2D, *p* = 0.0105). This correlation became even more evident when we considered the percentage of bone loss over time with respect to the individual patient, comparing the BMD value at T1 with the respective value at T0 (Figure 2E, *p* = 0.0021). Finally, we did not observe any correlation between the magnitude of BMD reduction and the gender (Figure 2F), or other factors, such as the level of specific biochemical markers at diagnosis, or the localization of the primary tumour, or HDCT treatment (data not shown). 

Finally, in EW patients only, when we compared the BMD value of the patient group at a one-year follow-up, we found that the use of radiotherapy had a negative impact on bone health (median BMD, 198.9 vs. 172.9 HU, *n* = 19 and 11, for not treated vs. treated patients, respectively, *p* = 0.015, Mann–Whitney U Test). However, this difference completely flattened out when we considered the reduction in BMD according to the individual patient’s trajectory of HU value (median Δ_T1–T0_, −30.8 vs. −27.9 HU, *n* = 19 and 11, for not treated vs. treated patients, respectively, Mann–Whitney U Test). 

### 3.4. BMD in Sarcoma Survivors

We continued the follow-up for an additional 4 years by collecting BMD data every year after chemotherapy. 

A total of 30 patients (20 OS, 10 EW) developed lung metastases during their follow-up, and 16 patients (14 OS, 2 EW) underwent thoracic surgery. In addition, 13 patients had a local recurrence (7 OS, 6 EW). Consequently, only 14 and 10 patients could be censored for OS and EW at the last follow-up, respectively (Table 1). However, we noticed that the BMD values at 4 and 5 years did not vary significantly (mean ± SEM, 220.5 ± 7.3 and 210.9 ± 10.1 HU, at T4 and T5, respectively, EW and OS patients merged, Figure 3A). Therefore, to retrieve clinical data from at least more than 15 patients per group and to study the outcome of bone health recovery in survivors, we used the difference in BMD between T4 vs. T0. At the first glance, an opposite trend between OS and EW patients was evident, as patients treated for EW did not recover at all (mean ± SEM, 195.5 ± 7.8 and 200.9 ± 9.4 HU, at T1 and T4, respectively). In fact, at 4 years, only 2 out of 16 patients (12.5%) regained BMD values equal to or higher than at the onset. The fraction of patients who at follow-up at T4 recovered a BMD similar to that observed at diagnosis was 26% for OS patients (7 out of 27). In Figure 3B, we also showed the differences between BMD values at T1 and at T4 compared to the difference between values at diagnosis and at T4 in the respective patient (Δ_Tx–T0_) to show that the results were almost the same, even when the intrapatient BMD decrease was considered. In contrast, in OS patients, BMD was almost completely restored (mean ± SEM, 200.5 ± 7.4 and 232.0 ± 9.6 HU, at T1 and T4, respectively) (Figure 3B, *p* < 0.0001). Notably, regardless of the type of bone sarcoma, the patient’s age at tumour diagnosis was significantly and inversely correlated with the patient’s BMD recovery (Figure 3C, *p* = 0.0183). Note that in our groups, EW patients are slightly older that OS patients (Table 1). Since the prepubertal and pubertal age during medication treatment may introduce a bias in the comparison of bone health outcome between OS and EW patients, we selected a subgroup of patients younger than 16 years. Indeed, the volumetric density of bone over time is highly dependent on the bone site [25]. In the spine, bone density peaks around the time of cessation of longitudinal growth, which corresponds to around 17 years in girls and 18 in males [26]. Again, we found a strong difference between EW and OS patients: BMD remained almost unchanged in EW patients from T1 to T4; whereas in OS, regardless the high rate of bone growth during chemotherapy, BMD had already almost reached the original value after 4 years (Figure 3D).

### 3.5. Severe Short-Term Bone Loss Is Predictive of a Low Likelihood of Long-Term Recovery in Bone Sarcoma Survivors

In order to study the risk of developing permanent bone loss based on specific clinical and personalized characteristics, it is crucial to consider the individual BMD trajectory. We thus assessed the ROI attenuation value of each individual over time and found that the most severe loss of BMD occurred shortly after one year, which is usually the time of completion of the treatment protocol (at T1, Figure 4A,B). 

In Figure 4A, patients with a Δ_T1–T0_ below the cut-off value were highlighted in red. From the graph, it is clear that patients marked in red did not return to the original BMD level, even 5 years after diagnosis. Sensitivity and sensibility were analysed and confirmed using the likelihood ratio test (predictive performance) (Table 2) and ROC analysis (Figure 4C, for OS this value was not significant). In conclusion, the Δ_T1–T0_ parameter was quite predictive, especially for EW patients.

## 4. Discussion

In this study, we opportunistically collected chest CT scan data to determine the disease-associated or drug-induced BMD loss in a homogeneous series of OS and EW patients by considering individual BMD trajectory. We also investigated whether, at one year after tumour onset, the extent of BMD loss was related to the persistence of abnormal low BMD in the long term. To achieve this goal, we used the HU value, obtained from CT scans and CT attenuation thresholds, as a surrogate marker of BMD, as previously validated by the comparison with DXA measurements, with the specific aim of distinguishing osteoporosis from non-osteoporosis [19,27]. We determined the mean HU of the anterior region of T12 ROI in a patient population that corresponded in age, sex, and survival to that reported in cancer statistics. 

At the onset, BMD values were not affected, as revealed by the comparison with a small group of non-oncologic subjects. Furthermore, in contrast to the data presented by Ruza et al. [14], we found no difference between OS and EW, nor any correlation of BMD with clinical features, such as tumour localization or volume, metastases at the onset, overall survival, or levels of biochemical markers that are associated with a worse prognosis, such as LDH and ALP. Finally, in patients with OS, we found that low BMD was correlated with high circulating levels of an inflammatory marker, CRP. Systemic inflammation is strongly associated with cancer development and progression [28], and serum CRP levels are directly associated with a worse outcome in many different malignancies, including bone sarcomas [29,30,31]. In addition, low-grade systemic inflammation affects physiological bone turnover and increases bone resorption through local activation of osteoclasts [32,33], and as we have previously shown, the most aggressive OS presents a high content of active osteoclasts [34]. In our series, we therefore hypothesized that in a number of patients, high CRP levels were a consequence of tumour-associated inflammation, which at the same time, also affected bone health and BMD. 

After one year, BMD tended to be lower than in the control group, corroborating the evidence of previous studies on a larger cohort [11,35], even as early as after only 7.8 months of treatment [12]. However, the difference with controls in our series was still not evident. In contrast, when we compared the BDM value at T1 with that at T0 in the same patient, the decrease became highly significant. This emphasizes how crucial it is to monitor bone loss by comparing values with the individual patient, rather than with reference values of the healthy population. The situation was even worse in metastatic patients. 

Loss of BMD at this stage of treatment could be due to several factors: first and foremost, the toxicity of polychemotherapy on bone, but also the reduced physical activity during treatment and frequent immobilization due to surgery, and the consequent decrease in muscle mass and vitamin D. In OS, two or more drugs are usually given together: the high-dose regimen of methotrexate, doxorubicin, and cisplatin (MAP regimen) is the most common first-line chemotherapy, although ifosfamide and etoposide are added in patients with poor response [36]. The most common sequence consists of four cycles of MAP (the start of the next cycle every 35 days and with a duration of 32 days), followed by two cycles of methotrexate alone and doxorubicin (the second cycle starts after 28 days and with a duration of 25 days), for a total duration of about 6–7 months [37,38]. For EW, multiagent regimens are preferred, based on vincristine, doxorubicin, cyclophosphamide/ifosfamide, and etoposide (VDC/IE), with interval-compressed regimens, alternating the combination of the two drugs, administered every 2–3 weeks, for seven plus seven cycles (each cycle starting after 28 days), for a total duration of around 13 months [37,38]. It is already widely known that high-dose intravenous chemotherapy regimens induce rapid bone loss due to their effects on the gonads [39,40]. In more detail, higher cumulative doses of methotrexate have been associated with a higher incidence of osteopenia and failure to regain normal BMD after completion of therapy [3,13,41,42]. In addition, it suppresses osteoblast activity, osteocyte viability, and stimulates osteoclast recruitment, resulting in decreased bone formation and increased bone resorption [12,43]. On the other hand, ifosfamide causes hypogonadism, resulting in decreased BMD, while cisplatin induces hypomagnesemia due to its renal toxicity. Magnesium deficiency also contributes to osteoporosis directly by acting on bone cells and altering crystal formation, and indirectly by affecting parathyroid hormone secretion and activity and promoting low-grade inflammation [12]. However, assessing the effect of each single drug on bone health is quite a challenging task because of the many confounding factors, such as age, treatment with multiple drugs, renal tubular dysfunction, presence of endocrinopathies, and drug-induced alteration of the gut microbiome, which in turn can induce osteoporosis [44], and to date, data on this subject are conflicting. 

Measuring BMD at clinical remission, Ruza et al. found that lumbar BMD in pubertal patients with OS and EW tended to be lower than that of patients who had completed pubertal development at the time of diagnosis [14]. However, in our patient population we found the opposite trend: at one year after the onset, younger patients had a smaller loss of mineral contents. We therefore hypothesized that this inconsistency was due to the use in previous studies of BMD reference standards that were based on the entire population, rather than on the same patient’s pretreatment BMD values, thus highlighting the advantage of intrapatient comparison for this type of analysis. This finding suggests that a favourable metabolic status and growth sprouting at an early age is an advantage for bone mass recovery in later life. In contrast, postpubertal patients missed the close window of high anabolic state and thus had a worse prognosis for bone health, and this should be considered by clinicians when assessing the risk of secondary osteoporosis in young patients with bone sarcomas.

We then considered the risk of failure to regain bone health even after several years (4–5 years after the onset) in survivors and its correlation with clinical characteristics. Again, our data were in contrast to previous findings showing that a high bone growth rate at prepubertal and pubertal age is a negative prognostic factor for long-term bone health outcome in patients with bone sarcoma undergoing chemotherapy [10]. In fact, in our cohort, as for T1, the reduction in BMD from T0 to T4 was inversely correlated with age, meaning that prepubertal and pubertal patients were at a lower risk of bone mineral loss than postpubertal patients. Analysing individual BMD trajectories over the 4-year follow-up, we also noticed that in both OS and EW groups, patients who had a greater loss of BMD at T1 matched with those subjects who did not recover bone health at the end of the follow-up. We then identified a cut-off value of BMD reduction at T1 that could predict the worst outcome, based on CT image analysis, and found a good degree of predictivity, with a cut-off corresponding to a loss of approximately more than 50 HU. Finally, with regard to the analysis of differences between OS and EW, our data are consistent with previous studies showing a higher incidence of osteopenia in EW patients [11,15,36,45]. We noted a clear asymmetry between OS and EW, with a worse course in patients with EW who usually have a longer treatment duration than patients with OS. In fact, in EW, the negative peak of BMD at one year did not return to initial levels, whereas in OS patients BMD usually recovered.

Our study has some limitations, including the small size of the OS and EW groups, the retrospective study design, and the lack of information on other markers/factors related to bone remodelling (e.g., physical activity, diet during therapy) that could reduce BMD. These, however, cannot be retrieved in a retrospective analysis. Furthermore, the number of cases was reduced to include only data from a single scanner and a single scanning protocol in order to limit the variability of HU measurements. For future studies, the bone density measurement protocol that relies on measuring data from a single CT slice, using an operator-chosen ROI [19], can be improved or overcome by increasing automation and including volumetric measurements, or by adopting asynchronous calibration protocols that have good reliability [46], or by using phantom-free approaches that rely on tissue-based density calibration, although these are associated with higher uncertainties [47]. 

## 5. Conclusions

In this work, we confirmed an increased risk of bone loss in bone sarcoma survivors, especially in EW patients, and provided novel findings that highlight the importance of studying individual trajectories to verify treatment-induced bone loss in each individual patient, according to a personalized approach. Indeed, the growth rate of bone volume has a defined and common pattern during human development, with the peak of bone growth occurring during puberty, which is also the most frequent period of bone sarcoma development. However, the rates of bone growth and BMD value in the spine at a specific age are strongly influenced by numerous factors that vary considerably among individuals, including race, gender, genetic factors, physical activity, calcium and protein intakes, weight, age of menarche, poor nutritional status, and a delayed onset of puberty [11,48].

This individual perspective paves the way for the identification of patients at risk of developing morbidities related to abnormal early bone loss, such as osteoporosis and increased fracture risk. This type of analysis can be performed by comparing CT analysis at the onset and at one-year follow-up. The use of the chest CT scan is already standard in these patients, and therefore does not require additional time, cost, or radiation exposure. This method may significantly impact on the long-term quality of life of bone sarcoma survivors: it allows for early preventive intervention aimed at avoiding the development of secondary osteoporosis, with specific recommendations, as suggested by the COG (e.g., vitamin D intake, adequate dietary calcium, promotion of regular physical activity with weight-bearing loads, or anticatabolic drugs such as bisphosphonates). Regarding the use of bisphosphonates, OS2006 and EuroEwing2012 clinical trials included the use of zoledronate in combination with chemotherapy [49,50]. However, the aim of these studies was to evaluate the effect of zoledronate on event-free survival, which shows no significant improvement for OS; and the analysis of the effect on the incidence of osteoporosis was postponed to a longer follow-up for forthcoming studies [49]. EuroEwing2012 results have not been published yet. For the future, it would be interesting to apply our method to such series.

In conclusion, the use of CT analysis to assess the extent of bone loss and validate the cut-off value of BMD loss at one-year follow-up to predict the risk of irreversible bone loss in patients with bone sarcoma deserves further investigation, including through prospective longitudinal ad hoc studies.

## Figures and Tables

**Figure 1 jcm-11-05412-f001:**
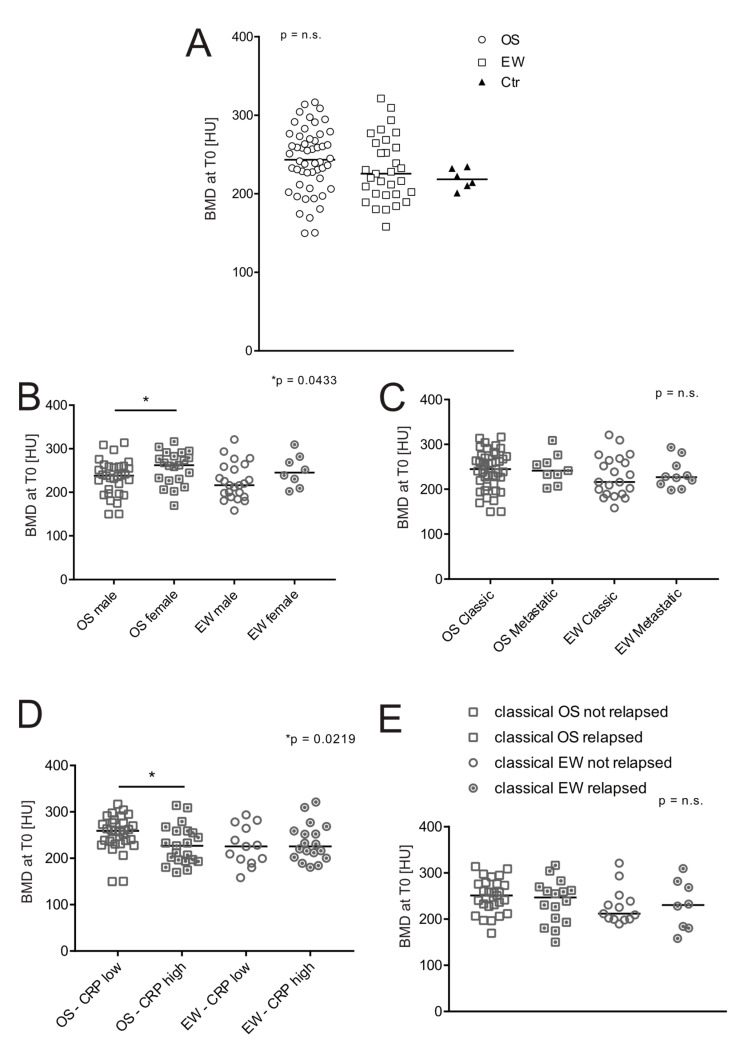
Effect of age, sex, and clinical features on BMD at the onset of patients with bone sarcoma. BMD was measured by ROI attenuation of T_12_ vertebrae assessed by CT in OS (*n* = 52), EW (*n* = 31), and control group (Ctr, *n* = 6). (**A**) Values and median of BMD values at the onset; (**B**) values shown in panel A, divided by sex (Mann–Whitney U test, * *p* < 0.05); (**C**) BMD values shown in panel A, divided by the presence/absence of clinically relevant metastases at diagnosis (Mann–Whitney U test); (**D**) BMD values shown in panel A, divided by low/high CRP values (CRP high > 0.5 mg/100 mL, Mann–Whitney U test, * *p* < 0.05); (**E**) BMD values shown in panel A and divided by the presence of relapse at follow-up (Mann–Whitney U test).

**Figure 2 jcm-11-05412-f002:**
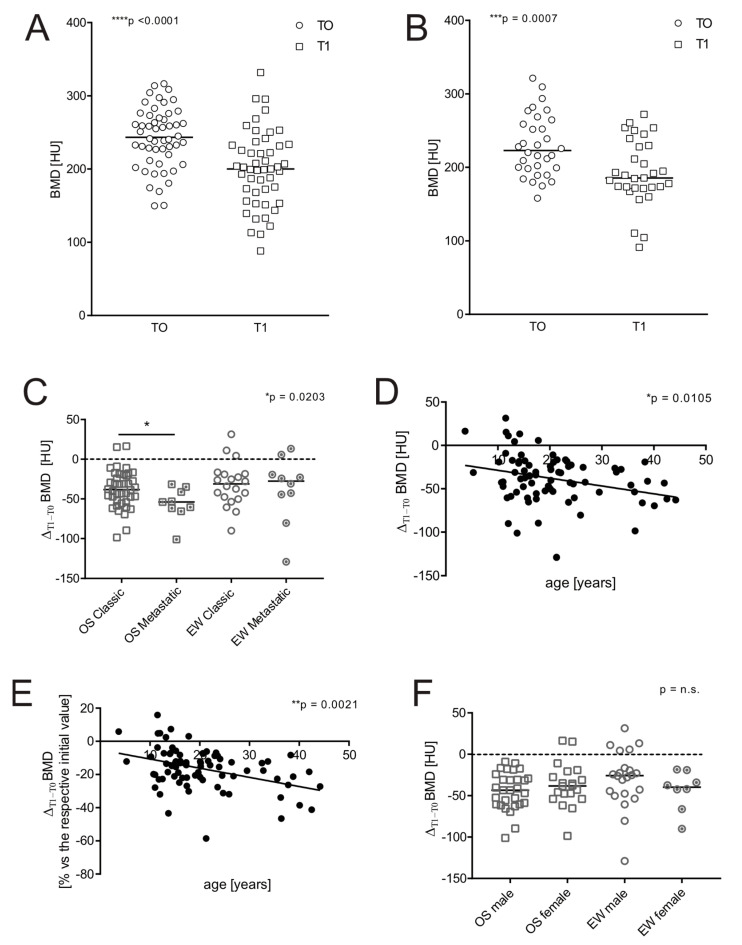
Effect of age, sex, and clinical features on BMD at 1 year after treatments of patients with bone sarcoma. In OS (*n* = 52), EW (*n* = 31) patients, immediately after completion of treatment protocols (around 1 year after diagnosis, T1), BMD was re-measured with the same method used at T0, by CT assessment. (**A**) Values and median of BMD at T0 vs. T1 of OS patients (Mann–Whitney U test, **** *p* < 0.0001); (**B**) values and median of BMD at T0 vs. T1 of OS patients (Mann–Whitney U test, *** *p* < 0.001); (**C**) values of Δ_T1–T0_ BMD (difference between BMD value at T1 respect to BMD value at T0) shown in panel B, divided by the presence/absence of clinically relevant metastases at diagnosis (Mann–Whitney U test, * *p* < 0.05); (**D**) correlation between Δ_T1–T0_ BMD values shown in panel B and age (OS and EW patients were grouped together, Spearman correlation test, * *p* < 0.05); (**E**) correlation between the percentage of BMD loss at T1, compared with the BMD value at T0 of the same patient and age (OS and EW patients were grouped together, Spearman correlation test, ** *p* < 0.005); (**F**); Δ_T1–T0_ BMD values and median shown in panel B, divided by gender (Mann–Whitney U test).

**Figure 3 jcm-11-05412-f003:**
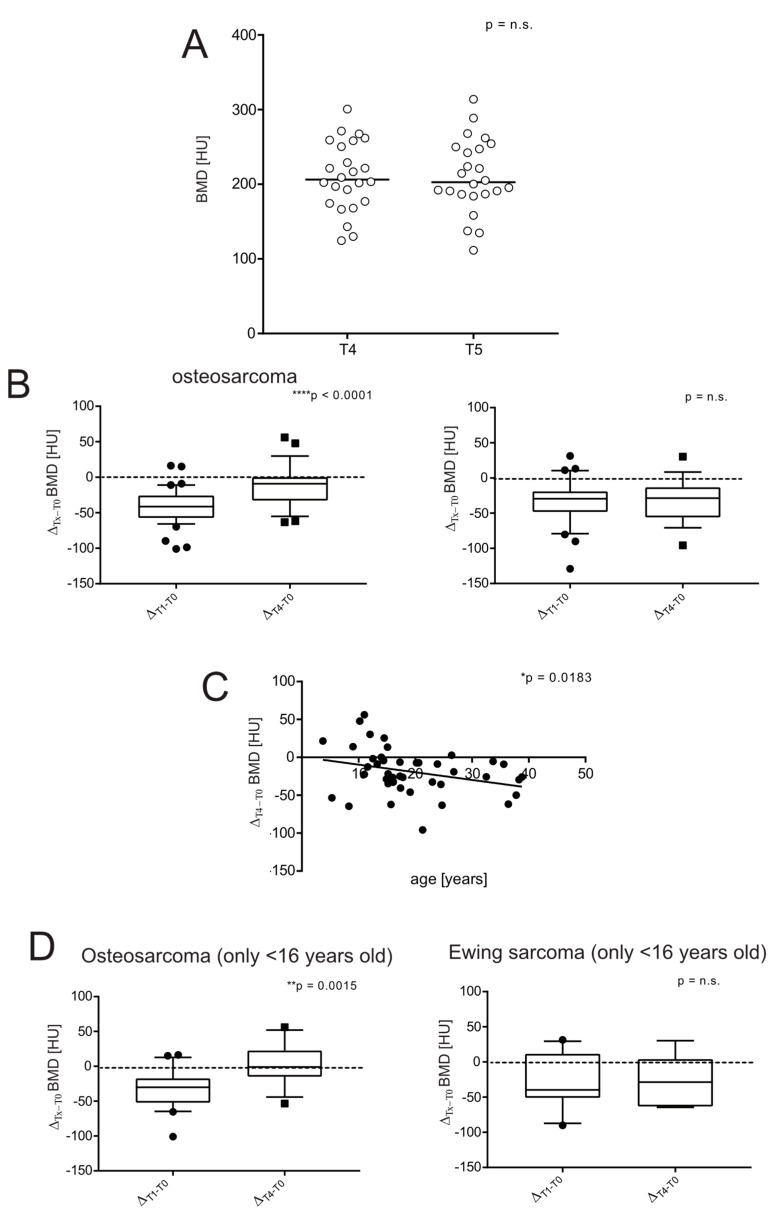
Long-term recovery of BMD: Ewing sarcoma had a worse prognosis. (**A**) BMD was measured by ROI attenuation level of T_12_ vertebrae as assessed by CT in OS and EW merged together, at time points T4 and T5 (*n* = 24, Wilcoxon paired test); (**B**) box plot of the difference of BMD values of Δ_T1–T0_ compared with the Δ_T4–T0_ difference in OS and EW (*n* = 49 at T0 and 27 at T1 for OS and *n* = 30 at T0 and 17 at T1 for EW, Mann–Whitney U test, **** *p* < 0.0001); (**C**) correlation between the different of BMD values at 4 years from BMD values at diagnosis (Δ_T4–T0_) and age at the onset (*n* = 43, Spearman correlation test, * *p* < 0.05); (**D**) box plot of Δ_T1–T0_ BMD values compared to Δ_T4–T0_ BMD in OS and EW who were less than 16 years old at the onset (*n* = 49 at T0 and 27 at T1 for OS and *n* = 30 at T0 and 17 at T1 for EW, Mann–Whitney U test, ** *p* < 0.001).

**Figure 4 jcm-11-05412-f004:**
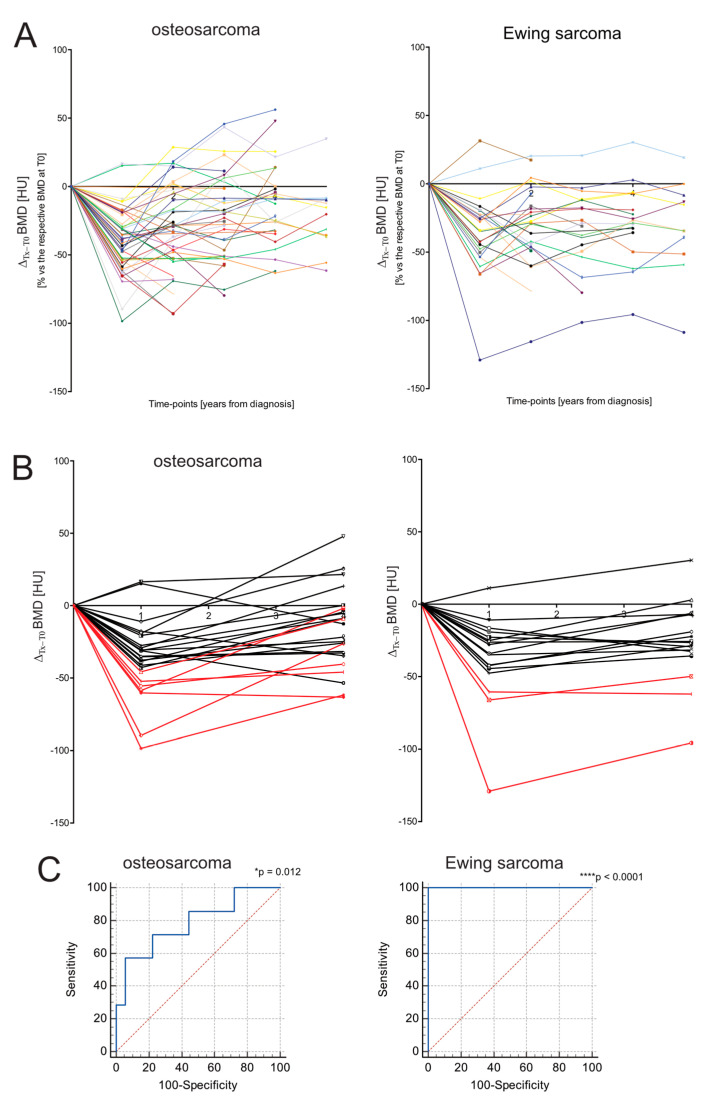
BMD trajectory at different follow-ups. (**A**) BMD was measured by ROI attenuation level of T_12_ vertebrae as assessed by CT in OS (*n* = 25), EW (*n* = 15) at different follow-ups. The trajectory of BDM values over time is expressed as % of decrease in BMD value between T0 and the different time points (Tx) in respect the initial BMD value of the respective patient. We also intended to use the extent of Δ_T1–T0_ BMD as a predictor inversely related to the ability of patients to restore their bone health. Each individual patient has a specific color. (**B**) BMD trajectory at different follow-ups by considering only T0, T1, and T4 to highlight the bone recovery or the loss of bone recovery by analysing individual trajectories. We used the 70° percentile of the Δ_T1–T0_ BMD (<−49.23 HU) as the cut-off, considering all patients with OS and EW (median range −27.83 HU, range −129.00 to 31.4). Red lines correspond to the trajectory of patients with a Δ_T1–T0_ BMD above the cut-off; (**C**) ROC curves for OS and EW to evaluate the predictive performance of the Δ_T1–T0_ BMD values. Recovery of bone health at 4 years after diagnosis in the individual patient was evaluated by using the Δ_T4–T0_ BMD value as a parameter (* *p* < 0.01, **** *p* < 0.0001).

**Table 1 jcm-11-05412-t001:** Characteristics of study participants. Alkaline phosphatase (ALP), lactate dehydrogenase (LDH), C-reactive protein (CRP).

Characteristics	OS (*n* = 52)	EW (*n* = 31)	Control (*n* = 6)
Sex	*n (%)*	*n (%)*	*n (%)*
Male	31 (59.6)	23 (74.2)	4 (66.7)
Female	21 (40.4)	8 (25.8)	2 (33.3)
Age (years)	*Median (range)*	*Median (range)*	*Median (range)*
Total	17.4 (3.7–44.2)	20.2 (8.4–38.3)	20.3 (8.8–23.6)
Male	19.1 (5.3–44.2)	20.2 (8.4–36.3)	20.3 (19.5–23.6)
Female	15.2 (3.7–36.4)	20.3 (11.0–38.3)	15.2 (8.8–21.7)
Localization			
Lower limbs	40	19	-
Axial	3	11	-
Upper limbs	9	2	-
Biochemical markers	*mean ± SEM*	*mean ± SEM*	-
ALP [U/L],	313.5 ± 65.3	126.3 ± 13.8	-
LDH [U/L]	461.6 ± 34.2	430.5 ± 27.1	-
CRP [mg/100 mL]	1.8 ± 0.7	3.3 ± 0.8	-
Metastatic at diagnosis			
N	9	10	
%	17%	32%	
Months from T0 (mean ± SEM) at T1	11.0 ± 0.4	11.3 ± 0.3	-
*n* of censored patients at T1	49	30	-
Months from T0 (mean ± SEM) at T2	23.1 ± 0.5	23.3 ± 1.0	-
*n* of censored patients at T2	37	21	-
Months from T0 (mean ± SEM) at T3	35.0 ± 0.6	34.4 ± 0.4	-
*n* of censored patients at T3	36	16	-
Months from T0 (mean ± SEM) at T4	47.6 ± 0.5	46.7 ± 0.7	-
*n* of censored patients at T4	28	16	-
Months from T0 (mean ± SEM) at T5	58.9 ± 0.9	58.5 ± 10.1	-
*n* of censored patients at T5	14	10	-

**Table 2 jcm-11-05412-t002:** ROC curve parameters and likelihood ratio test for the evaluation of the predictive performance of bone loss, as measured by QCT (Δ_T0–T1_), for low probability of long-term bone health recovery in patients with OS and EW, at 4 years (* *p* < 0.05, **** *p* < 0.0001).

Prognostic Performance	OS	EW
Sample size		
Total	25	15
Positive group	7 (28%)	3 (20%)
Negative group	18 (72%)	12 (80%)
Area under the ROC curve	0.786 ± 0.114 (fair)	1 ± 0 (excellent)
95% Confidence interval	0.577 to 0.923	0.782 to 1.00
*p* value	0.012 (*)	<0.0001 (****)
Sensitivity		
Value	57.14%	100%
95% CI	18.41% to 90.10%	29.24% to 100%
Specificity		
Value	83.33%	83.33%
95% CI	58.58% to 96.42%	51.59% to 97.91%
Prevalence		
Value	28%	20%
95% CI	12.07% to 49.39%	4.33% to 48.09%
Accuracy		
Value	76.00%	86.67%
95% CI	54.87% to 90.64%	59.54% to 98.34%
Positive predictive value		
Value	57.14%	60.00%
95% CI	28.33% to 81.81%	29.74% to 84.17%
Negative predictive value		
Value	83.33%	100%
95% CI	67.47% to 92.34%	-
Likelihood ratio test +		
Value	3.43	6.00
95% CI	1.02 to 11.57	1.69 to 21.26
Likelihood ratio test −		
Value	0.51	0.00
95% CI	0.21 to 1.24	

## Data Availability

Data are available to academic investigators from the authors upon reasonable request.

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
