# Peer review of "Individual Trajectories of Bone Mineral Density Reveal Persistent Bone Loss in Bone Sarcoma Patients: A Retrospective Study"

_jcm, 2022, doi:10.3390/jcm11185412_

Round 1

Reviewer 1 Report

Interesting study on bone mineral density after multimodal treatment of bone sarcoma in children and young adults. 

The methodology is robust and the results are well described and analyzed. 

The discussion is interesting and the conclusion are in accordance with the results. 

Suggestions: 

In the Subjects and Methods section: can the authors indicate whether some patients included received bisphosphonates during or after treatment, according to osteosarcoma therapeutic protocols in the late 2000’s (OS 2006 protocol). 

In the result section:  figures 3A and 3B not very informative. The comparison between T0 and T1 should be shown in figure 3 (could be added in 3A and 3B). 

In the discussion section: one or 2 phrases should indicate the effect of bisphosphonates during and after treatment in OS and EW patients on cancer treatment as the results of OS2006 and EuroEWING2012 have been published, and if those can be a option to prevent bone loss induced by chemotherapy.  

Detailed analysis : 

Introduction : 

The problem is well defined: lower BMD (secondary osteoporosis) in cancer survivors can be responsible for late complications. It is even more relevant in pediatric patients and young adults, such as bone sarcoma patients, because of longer life expectancy. But is has not been established yet how to identify patients in whom BMD is permanently compromised.   

Objectives are given at the end of the introduction

  • Analyze individual trajectories to ascertain the effect of chemotherapy on bone health
  • Identify patients at higher risk of developing secondary osteoporosis

The study design is also given:  retrospective case series, osteosarcoma and Ewing sarcoma patients, BMD data before treatment and up to 5 years of follow up to.

Subjects and Methods:

Patients: inclusion and exclusion criteria well described.

3 groups of patients: 52 OS, 31 EW, 6 controls.

Measurement of BMD in T12 vertebrae on chest CT according to validated method on Hounsfield units (with checked reference) at diagnosis and every year up to 5 years of follow up.

Other data: patients characteristics, laboratory parameters (CRP, ALP, LDH).

Statistics: use of non-parametric tests such as Mann Whitney, Wilcoxon, Spearman Rank test. ROC curves, sensitivity and specificity, likelihood ratios.

Results: 

Studied population : consistent with literature

BMD at onset : female OS had higher BMD than male OS, higher CRP correlated with lower BMD in OS patients.

BMD at completion of chemotherapy :

  • OS and EW patients had significant reduction of BMD.
  • Lung metastases at onset in OS patients : greater loss of BMD.
  • BMD loss inversely correlated with patient age.
  • In EW patients : radiotherapy had a negative impact on bone health.

BMD in sarcoma survivors : 14 OS and 10 EW

  • EW did not recover at all
  • Patients age at tumor diagnosis was significantly and inversely correlated with patient’s BMD recovery.

Severe short term bone loss is predictive of a low likelihood of long term recovery in bone sarcoma survivors

  • Most severe loss of BMD : occurred shortly after one year.
  • QCT (ΔT1-T0) parameter : good predictor of long term compromised BMD, especially in EW patients.

Discussion:

Good quality discussion on the results, compared with the literature.

Conclusion: no comment. 

Author Response

Interesting study on bone mineral density after multimodal treatment of bone sarcoma in children and young adults. 

The methodology is robust and the results are well described and analyzed. 

The discussion is interesting and the conclusion are in accordance with the results. 

Suggestions: 

In the Subjects and Methods section: can the authors indicate whether some patients included received bisphosphonates during or after treatment, according to osteosarcoma therapeutic protocols in the late 2000’s (OS 2006 protocol). 

> ‘Patients included in this study were not enrolled in the randomized EuroEWING2012 (clinicaltrialsregister.eu, EudraCT Number: 2012-002107-17) and OS2006 (clinicaltrial.gov) protocols in which zoledronate was evaluated for its presumptive anticancer activity’. We added also a comment in the ‘Subject and Method section’, as requested.

In the result section:  figures 3A and 3B not very informative.

> As suggested by the reviewer, Fig. 3A and B were withdrawn from Fig. 3.

The comparison between T0 and T1 should be shown in figure 3 (could be added in 3A and 3B). 

> As suggested by the reviewer we added the comparison of row data of BMD at T0 vs T1 for both OS and EW in panels A and B of Fig. 3.

In the discussion section: one or 2 phrases should indicate the effect of bisphosphonates during and after treatment in OS and EW patients on cancer treatment as the results of OS2006 and EuroEWING2012 have been published, and if those can be a option to prevent bone loss induced by chemotherapy.  

> As suggested by the reviewer we added a comment relative to the results of the clinical trials OS2006 and EuroEWING2012 in the final section of the manuscript, as follows: “Regarding the use of bisphosphonate, OS2006 and EuroEwing2012 clinical trials included the use of zoledronate in combination with chemotherapy [49,50]. However, the aim of these studies was to evaluate the effect of zoledronate on event-free survival, which shows no significant improvement for OS, and the analysis of the effect on the incidence of osteoporosis was postponed to longer follow-up for forthcoming studies [49]. EuroEwing2012 results have not been published yet. For the future, it would be interesting to apply our method to such series.”.

Detailed analysis : 

Introduction : 

The problem is well defined: lower BMD (secondary osteoporosis) in cancer survivors can be responsible for late complications. It is even more relevant in pediatric patients and young adults, such as bone sarcoma patients, because of longer life expectancy. But is has not been established yet how to identify patients in whom BMD is permanently compromised.   

Objectives are given at the end of the introduction

  • Analyze individual trajectories to ascertain the effect of chemotherapy on bone health
  • Identify patients at higher risk of developing secondary osteoporosis

The study design is also given:  retrospective case series, osteosarcoma and Ewing sarcoma patients, BMD data before treatment and up to 5 years of follow up to.

Subjects and Methods:

Patients: inclusion and exclusion criteria well described.

3 groups of patients: 52 OS, 31 EW, 6 controls.

Measurement of BMD in T12 vertebrae on chest CT according to validated method on Hounsfield units (with checked reference) at diagnosis and every year up to 5 years of follow up.

Other data: patients characteristics, laboratory parameters (CRP, ALP, LDH).

Statistics: use of non-parametric tests such as Mann Whitney, Wilcoxon, Spearman Rank test. ROC curves, sensitivity and specificity, likelihood ratios.

Results: 

Studied population : consistent with literature

BMD at onset : female OS had higher BMD than male OS, higher CRP correlated with lower BMD in OS patients.

BMD at completion of chemotherapy :

  • OS and EW patients had significant reduction of BMD.
  • Lung metastases at onset in OS patients : greater loss of BMD.
  • BMD loss inversely correlated with patient age.
  • In EW patients : radiotherapy had a negative impact on bone health.

BMD in sarcoma survivors : 14 OS and 10 EW

  • EW did not recover at all
  • Patients age at tumor diagnosis was significantly and inversely correlated with patient’s BMD recovery.

Severe short term bone loss is predictive of a low likelihood of long term recovery in bone sarcoma survivors

  • Most severe loss of BMD : occurred shortly after one year.
  • QCT (ΔT1-T0) parameter : good predictor of long term compromised BMD, especially in EW patients.

Discussion:

Good quality discussion on the results, compared with the literature.

Conclusion: no comment. 

> Thank you for your comments and your careful revision

Reviewer 2 Report

This is a retrospective study on surviving children and adolescents suffering from osteosarcoma and Ewing sarcoma, 2 malign diseases with limited life expectancies. BMD at diagnosis and in the following years has been generated and analyzed from at Th 12 from routine CT and expressed as Hounsfield units. A method which also my group has applied years ago in patients with COPD.

In my view this is an excellent study on a very sensible topic. The results derived on the basis of a very sophistic methodology are very important since the treatment options become more and more efficient leading to better quality of life and life expectancy in these young people. 

The introduction is giving a excellent overview on the topic, the methodology is reproducable , the results are presented on an interesting way. The discussion is instructive. The limitations of the study are as excpected, however the given aims are clear and unmistakable presented.

In summary congratulations to the authors         

Author Response

Thank you for your kind appreciation for the manuscript. It is really rewarding.

Reviewer 3 Report

The authors made a comprehensive analysis of the performance of the bone mineral density of a cohort of osteosarcoma and Ewing sarcoma patients.

The study has many tiny graphics and is sometimes difficult to interpret in a printed manuscript. I'm sure the authors could reduce de number of illustrations. I'm sure that figure 1 can be suppressed without compromising the paper quality.

On the other hand, I missed better information related to the tumor location. For example, tumors located in lower limbs, including pelvic tumors, can negatively affect BMD because those patients will take much more time to recover than patients in upper limb tumors. Those patients will spend much more time in bed. I know the authors described that there was not a significant difference regarding tumor location. Still, I would like to be sure the authors made a specific calculation comparing the BMD of patients with lower limbs (including pelvic) tumors with patients with upper limb tumors. 

Another potential bias is the type of reconstruction. For example, amputation and Prothesis usually have faster recoveries than biological reconstructions.

The results presentation has some discussion. I believe the authors could reduce the size of the paper by eradicating those comments from the results.

Please double-check the figure 2 legend. There is an inversion in the graphics C and D.

Author Response

The authors made a comprehensive analysis of the performance of the bone mineral density of a cohort of osteosarcoma and Ewing sarcoma patients.

The study has many tiny graphics and is sometimes difficult to interpret in a printed manuscript. I'm sure the authors could reduce de number of illustrations. I'm sure that figure 1 can be suppressed without compromising the paper quality.

> As suggested by the reviewer we have withdrawn Fig. 1 and enlarged the size of the other figures.

On the other hand, I missed better information related to the tumor location. For example, tumors located in lower limbs, including pelvic tumors, can negatively affect BMD because those patients will take much more time to recover than patients in upper limb tumors. Those patients will spend much more time in bed. I know the authors described that there was not a significant difference regarding tumor location. Still, I would like to be sure the authors made a specific calculation comparing the BMD of patients with lower limbs (including pelvic) tumors with patients with upper limb tumors.

> As requested by the reviewer, we repeated the analysis. We identified two groups of patients, as suggested, one group with the upper limb tumour and the other with the lower limb (including the pelvic tumour).  The reviewer can find the results in the "Figure for the reviewer". As we did not have a sufficient number of cases at T4, to perform a statistical analysis of bone loss compared to T0 (at T4 we could only recover the HU value for two patients in the group with upper limb tumour), we performed the analysis of T3 compared to T0 by merging the OS and EW cases. At both T1 and T3, we did not observe any difference between the lower and upper limb groups (Mann Whitney U-test, two-tailed). One possible explanation is that a few months after surgery, the weight bearing usually returns to normal, especially in young patients who are growing subjects with high bone remodeling activity. The follow-up we observed was after several months/years, so there is probably no long-term effect due to the increased time spent in bed immediately after surgery.

Another potential bias is the type of reconstruction. For example, amputation and Prothesis usually have faster recoveries than biological reconstructions.

> See the answer above.

The results presentation has some discussion. I believe the authors could reduce the size of the paper by eradicating those comments from the results.

> As suggested by the reviewer, we have reduced the text of the 'results section' by removing discussion or conclusion texts in this section.

Please double-check the figure 2 legend. There is an inversion in the graphics C and D.

> We revised and corrected the figure legend of Fig. 1 (namely Fig. 2), as suggested.